# Predicting and identifying factors associated with undernutrition among children under five years in Ghana using machine learning algorithms

Eric Komla Anku[1☯]*, Henry Ofori Duah[2☯]

1 Dietherapy and Nutrition, Cape Coast Teaching Hospital, Cape Coast, Ghana, 2 University of Cincinnati College of Nursing, Cincinnati, Ohio, United States of America

☯ These authors contributed equally to this work.
* ankueric1@gmail.com

## Abstract

### Background

Undernutrition among children under the age of five is a major public health concern, especially in developing countries. This study aimed to use machine learning (ML) algorithms to predict undernutrition and identify its associated factors.

### Methods

Secondary data analysis of the 2017 Multiple Indicator Cluster Survey (MICS) was performed using R and Python. The main outcomes of interest were undernutrition (stunting: height-for-age (HAZ) < -2 SD; wasting: weight-for-height (WHZ) < -2 SD; and underweight: weight-for-age (WAZ) < -2 SD). Seven ML algorithms were trained and tested: linear discriminant analysis (LDA), logistic model, support vector machine (SVM), random forest (RF), least absolute shrinkage and selection operator (LASSO), ridge regression, and extreme gradient boosting (XGBoost). The ML models were evaluated using the accuracy, confusion matrix, and area under the curve (AUC) receiver operating characteristics (ROC).

### Results

In total, 8564 children were included in the final analysis. The average age of the children was 926 days, and the majority were females. The weighted prevalence rates of stunting, wasting, and underweight were 17%, 7%, and 12%, respectively. The accuracies of all the ML models for wasting were (LDA: 84%; Logistic: 95%; SVM: 92%; RF: 94%; LASSO: 96%; Ridge: 84%, XGBoost: 98%), stunting (LDA: 86%; Logistic: 86%; SVM: 98%; RF: 88%; LASSO: 86%; Ridge: 86%, XGBoost: 98%), and for underweight were (LDA: 90%; Logistic: 92%; SVM: 98%; RF: 89%; LASSO: 92%; Ridge: 88%, XGBoost: 98%). The AUC values of the wasting models were (LDA: 99%; Logistic: 100%; SVM: 72%; RF: 94%; LASSO: 99%; Ridge: 59%, XGBoost: 100%), for stunting were (LDA: 89%; Logistic: 90%; SVM: 100%; RF: 92%; LASSO: 90%; Ridge: 89%, XGBoost: 100%), and for underweight were (LDA:

**Data Availability Statement:** This study was based on a publicly available dataset with no personal identifiers, and is freely available upon request from the Ghana Statistical Service website (https://

www.statsghana.gov.gh/gssdatadownloadspage.
php) and the MICS website (https://mics.unicef.
org/surveys) after an online registration. The code
used for the analysis can be found in this
repository (https://github.com/KomlaRD/machine_
learning_undernutrition).

**Funding:** The author(s) received no specific
funding for this work.

**Competing interests:** The authors have declared
that no competing interests exist.

95%; Logistic: 96%; SVM: 100%; RF: 94%; LASSO: 96%; Ridge: 82%, XGBoost: 82%).
Age, weight, length/height, sex, region of residence and ethnicity were important predictors
of wasting, stunting and underweight.

## Conclusion

The XGBoost model was the best model for predicting wasting, stunting, and underweight.
The findings showed that different ML algorithms could be useful for predicting undernutri-
tion and identifying important predictors for targeted interventions among children under five
years in Ghana.

## Introduction

Undernutrition (wasting, stunting, and underweight) among children under five years of
age is a public health issue with serious implications [1]. Undernutrition contributes to
nearly half of the mortality in children under five years of age, with the highest burden in
low- and middle-income countries [2, 3]. The burden of undernutrition is not limited to the
clinical or socioeconomic outcomes [3]. Global statistics indicate a decreasing trend in
undernutrition; however, it remains high in developing countries [4], including Ghana,
especially for stunting.

The global prevalence of stunting among children under five is 22%, and wasting is 7%
according to The United Nations Children's Fund (UNICEF) [2]. Ghana has made strides to
reduce the rate of undernutrition among children under five years old. Prevalence rates from
the 2014 Ghana Demographic and Health Survey (GDHS) reported stunting at 19%, wasting
at 5%, and underweight at 11% [5], with the highest burden in the northern regions, similar to
a study conducted in Ethiopia [6]. The rates of stunting, wasting, and underweight were 28%,
9%, and 14%, respectively in 2008 [7]. Stunting, wasting, and underweight were reported to be
18%, 7%, and 13%, respectively, in a recent Multiple Indicator Cluster Survey [8]. The trends
show a decrease in the rates of stunting but not wasting or underweight.

The use of machine learning (ML) algorithms is key to identifying the factors associated
with undernutrition and driving decisions to reduce it. Previous studies have relied on logistic
regression to determine the factors associated with undernutrition [9, 10], which may not
always be sufficient to identify patterns in data [11]. Machine learning approaches have shown
promising outcomes in identifying factors associated with undernutrition, including previ-
ously undiscovered factors [1]. Several studies have been conducted in Bangladesh [10, 12, 13],
Ethiopia [1, 6], and India [14, 15] and have shown the usefulness of machine learning algo-
rithms. Factors predicted to be significantly associated with undernutrition vary based on loca-
tion, including time to water source, anaemia history, child age > 30 months, low birth weight
and maternal underweight [1], urban-rural settlement, literacy factors of parents, and place of
residence [6, 12].

To the best of our knowledge, evidence on machine learning algorithms and undernutri-
tion among children under five years of age in Ghana is limited. Thus, this proof-of-concept
study aimed to provide evidence for the use of machine learning algorithms to predict
undernutrition among children under five years of age in Ghana and to identify associated
factors.

## Methods

### Data source

Data from the 2017 Multiple Indicator Cluster Survey (MICS) was used in this study. The MICS survey was conducted from October 2017 to January 2018. The original dataset for children under five years of age had 8906 data points. Data collection was performed in ten 10 administrative regions of Ghana. In each administrative region, the main sampling units were rural and urban areas. Thereafter, two-stage sampling was used to select households for the interviews.

### Data preparation

Data were downloaded from the MICS website. Data wrangling was completed in R version 4.3.0 using *tidyverse* packages [16]. Study variables and measurements

**Outcome.** The main outcomes of interest were three nutritional indicators in children under five years of age: stunting, wasting, and underweight. The z-scores of the anthropometric measures were used to assess nutritional indicators: weight-for-height (wasting), height-for-age (stunting), and weight-for-age (underweight) were used to evaluate nutritional status. Based on the World Health Organization (WHO) criteria, a child was classified as wasted if the weight-for-height z-scores were < -2 SD, stunted if the height-for-age z-scores were < -2 SD, and underweight if the weight-for-age z-scores were < -2SD. The three nutritional outcomes were coded as 0 for the absence of nutritional indicators and 1 for the presence of nutritional indicators. Therefore, children who were wasted, stunted, and underweight were all given a code of 1; otherwise, they were coded as 0 for normal indicators.

### Covariates

We considered a set of covariates to be predictors of malnutrition in Ghana. We included a set of variables based on the literature and availability in the dataset, excluding those with missing cases of > 50%. The following covariates were considered: age, sex, region, area, length/height, weight, child ill with cough for two weeks, child ill with fever for two weeks, child had diarrhoea for two weeks, health insurance, mother's educational level, ethnicity, and combined wealth score.

### Analytic strategy

R and Python programming languages were used for the analysis. The *tidyverse* packages [16] from R were used for data wrangling and the *scikit-learn* package [17] from Python for machine learning. The *survey* [18] package in Rwas used to create a survey object that accounted for the primary sampling unit, stratification, and sample weight of the in the univariate and bivariate analysis. Seven ML algorithms (Linear Discriminant Analysis (LDA), Logistic Model, Support Vector Machine (SVM), Random Forest, least absolute shrinkage, selection operator (LASSO) regression, Ridge Regression, and Extreme Gradient Boosting (XGBoost)) were trained for each nutritional indicator. A summary of each algorithm is provided below:

**Linear discriminant analysis (LDA).** Linear discriminant analysis (LDA) is a dimensionality reduction technique that is used for classification. It is a supervised machine learning technique used to find a linear combination of features for the optimal classification of known groups. The goal of LDA is to find the linear axis in the dimensional space that maximizes the distance between the means of classes while minimizing the variability within the classes to ensure the optimal separation of classes [19].

**Logistic regression.** Logistic regression is a supervised machine learning technique used to classify binary outcomes or classes based on prediction probabilities. Logistic regression is similar to a linear regression model, but it uses a complex cost function called the 'Sigmoid function' or the 'logistic function' instead of the linear function in linear regression [20]. The logistic function helps transform predicted probabilities to lie between zero and one [20]. A decision boundary (threshold) is then used to create an optimal classification based on the probability score. Therefore, when the predicted probability is above the threshold, it is grouped into one class, and those with predicted probabilities below the threshold are grouped into a separate class [20].

**Support vector machine (SVM).** Support vector machine is a supervised ML technique used for classification and regression. The goal of SVM is to identify a hyperplane that ensures the optimal separation of classes. The support vectors represent the data points that are nearest to the other sides of the hyperplane, which are critical for their removal to change the position of the dividing hyperplane [21]. The margin is the distance between the hyperplane and nearest data point from each side. The goal of SVM is to select the ideal hyperplane that has the maximum margin between the hyperplane and any data point in the training data to ensure good separation. For complex classification, SVM can use 3-Dimensional (3-D) space to enhance separation through the process of kernelling [21].

**Random forest.** Random forest (RF) is a supervised ML technique that leverages insights from several decision trees for classification. Unlike a decision tree, which uses only one tree to make predictions, a random forest fits several uncorrelated classification trees and uses the average of all individual trees to ensure an improved prediction accuracy compared with individual trees. Random forest uses bootstrap aggregation or bagging to select a random sample (with replacement) of the training dataset for each decision tree of the decision trees used in the RF. Moreover, the splitting of nodes in the RF model is based on a random set of features for each tree. Therefore, the features of each tree could vary from one another. The bagging process and variations in feature selection for each tree contribute to the robustness of RF prediction accuracy [22].

**Least absolute shrinkage and selection operator (LASSO) regression.** LASSO regression is a supervised ML technique that employs L1 regularization to control overfitting of data during regression. The L1 regularization method applies a penalty to the magnitude of the coefficients associated with each independent variable in the model. The penalty shrinks the less important coefficients towards zero, essentially eliminating them from the model. The tuning parameter ($\lambda$) is used to control the strength of the penalty in LASSO [23].

**Ridge regression.** Ridge regression is a supervised ML technique that uses L2 regularisation to overcome overfitting of the training data during regression. L2 regularisation in ridge regression penalises the loss function by adding the squared absolute values of the magnitudes of the coefficients as penalty terms [23].

*XGBoost*. XGBoost is a scalable tree-boosting system designed to improve the performance of machine learning models. It combines the strengths of the gradient boosting and regularisation techniques. XGBoost uses a novel regularisation term that penalises complex models and allows better control over model complexity. It employs a parallel and distributed computing framework to efficiently handle large-scale datasets [24].

## ML approach

We trained the algorithms to identify features that predict nutritional indicators in children under five years in Ghana. First, the data were divided into training and test datasets. We used 70% and 30% for the training and testing datasets, respectively. The training dataset for the

wasting was oversampled to deal with class imbalance. We trained all seven ML algorithms, including the LDA, SVM, logistic, RF, LASSO, ridge models, and XGBoost on the training set separately for wasting, stunting, and underweight. We used a 5-fold cross validation to tune the hyperparameters of the models. Moreover, the same random seed was applied to ensure that the same training and validation sets were obtained while training different ML algorithms.

## Algorithm evaluation

We evaluated the accuracy of the models on the test dataset using a confusion matrix and area under the curve receiver operating characteristic (AUC-ROC) plots. The accuracy, sensitivity, and specificity of the models were evaluated using a confusion matrix. Assuming a confusion matrix for a standard binary classifier, the four possible outcomes are listed in Table 1.

**Accuracy.**   The accuracy is the ratio of the total number of correct predictions (TP + TN) to the total number of predictions (TP + TN + FP + FN). Therefore, the accuracy is an estimate of the overall ability of the classifier to make correct predictions. Mathematically,

**Accuracy** = TP + TN / TP + TN + FP + FN

**Sensitivity.**   The sensitivity of a classifier is defined as the ratio of the number of positive cases correctly classified by the model to the total number of positive cases. Sensitivity refers to the ability of a classifier to designate an individual with disease as positive. A highly sensitive classifier has a small proportion of false negatives, resulting in only a few missed cases. Mathematically,

**Sensitivity** = TP / TP + FN

**Specificity.**   The specificity of the classifier is the ratio of the number of negative cases correctly classified by the model to the total number of negative cases. Specificity refers to the ability of a classifier to designate an individual without disease as negative. A highly specific classifier has a small proportion of false positives, resulting in only a few noncases being incorrectly diagnosed. Mathematically,

**Specificity** = TN / TN + FP

**Area under the curve receiver operating characteristics (AUC-ROC).**   The receiver operating characteristics (ROC) curve is a unit square plot that shows the diagnostic ability of a classifier. It is produced by plotting the true positive rate (sensitivity) and false positive rate (1-specificity). The area under the curve (AUC) estimates the area under the ROC curve by computing the aggregate performance of a classifier under different thresholds. The AUC values range from 0 to 1. An AUC value of 0.5 implies that the classifier has no ability to discriminate and is no better than chance. An AUC of 1 implies that the classifier has a perfect discrimination. Therefore, the closer the AUC value of a classifier is to 1, the better is the classifier, which provides a basis for evaluating and comparing the discrimination abilities of competing classifiers.

**Variable importance.**   We estimated the variable importance of the algorithms using Python. Variables with high values implied that they made an important contribution to the overall model accuracy.

**Table 1. A sample confusion matrix of binary classifier.**

|  |  | Predicted | |
|---|---|---|---|
|  |  | Positive | Negative |
| **Observed** | **Positive** | True Positive (TP) | False Negative (FN) |
|  | **Negative** | False positive (FP) | True Negative (TN) |

### Ethical considerations

Ethical approval was not required for this secondary analysis. Verbal consent was obtained from each adult participant and children aged between 15 and 17 years during the primary data collection. For the younger children, consent was obtained from their parents or caregivers. Participants were informed of their right to voluntary participation, confidentiality, and anonymity of the information obtained. Personal identifiable information was removed from the data set.

## Results

### Sample characteristics

A total of 8564 children under five years of age were included in the analysis. The mean age of the patients was 926 days; 51% were females, and 57% resided in urban areas. Thirty-seven percent of the mothers had attained junior secondary education Table 2.

### Weighted prevalence of nutritional indicators and associated factors

The weighted prevalence rates of wasting, stunting, and underweight were 7%,17% and 12%, respectively (Table 2). The children who were wasted were younger than the non-wasted children ($p < 0.001$). Most of the children who wasted were males (57%) rather than females (43%) ($p = 0.029$). Children who were wasted had significantly lower current weight ($8.1 \pm 2.6$ kg) than children without wasting ($11.8 \pm 3.4$ kg), $p < 0.001$. No differences were observed in the burden of wasting in terms of region of residence, area of residence, whether a child had diarrhoea or not, mothers' educational level and the combine wealth score of the household.

Majority of the children who were stunted were males (55%) than females (45%), $p = 0.007$. A higher proportion of children who were stunted were from rural areas (67%) ($p < 0.001$). Children who were stunted were older than those who were not stunted ($p < 0.001$). There was a significant association between stunting and the following variables: region of residence, current weight and height, child who had fever or diarrhoea 2 weeks ago, the educational level of the mother, ethnicity, and combined wealth score ($p < 0.05$).

There was a significant association between underweight and the following variables: age of the child, sex of the child, region of residence, current weight and height/length of the child, child ill with fever two weeks ago, child had diarrhoea two weeks ago, whether child had health insurance, ethnicity, and combined wealth score ($p < 0.05$) (Table 3).

### Predictive algorithms for child undernutrition indicators and associated receiver operator characteristics on the test data

**Wasting.**   The under-five wasting prediction accuracies were found to be high for all algorithms on the test data (84–98%). Accuracy of the XGBoost model was highest (98%) for predicting wasting followed by LASSO, Logistic model and RF (Table 4). The sensitivity of LDA, Logistic model, LASSO and Ridge were higher compared to the other ML models. RF, however, was the highest predictive model in terms of specificity followed by XGBoost and SVM. Based on AUC values, XGBoost and Logistic model had the highest performance (Fig 1).

**Stunting.**   The accuracy of stunting prediction for all algorithms ranged between 86% and 98% for the test data. XGBoost and SVM had the highest value in terms of accuracy, with an optimal balance of sensitivity and specificity relative to the other models (Table 4). The best model for predicting stunting in children under five years based on ROC-AUC values were XGBoost and SVM with an AUC of 100%, which indicates that they had the greatest discrimination compared to the other models (Fig 2).

**Table 2. Sociodemographic and anthropometric characteristics of children.**

| Characteristic | N = 8,564[1] |
|---|---|
| **Age (days)** | 926 (520) |
| **Sex** | |
| Female | 4,331 (51%) |
| Male | 4,233 (49%) |
| **Region** | |
| Ashanti | 2,055 (24%) |
| Brong Ahafo | 803 (9.4%) |
| Central | 907 (11%) |
| Eastern | 890 (10%) |
| Greater Accra | 826 (9.7%) |
| Northern | 996 (12%) |
| Upper East | 279 (3.3%) |
| Upper West | 209 (2.4%) |
| Volta | 686 (8.0%) |
| Western | 911 (11%) |
| **Area** | |
| Rural | 4,866 (57%) |
| Urban | 3,698 (43%) |
| **Length/Height (cm)** | 86 (14) |
| **Weight (kg)** | 11.6 (3.5) |
| **Child ill with cough** | |
| No | 5966 (70%) |
| Yes | 2,598 (30%) |
| **Child ill with fever** | |
| No | 6358 |
| Yes | 2,206 (26%) |
| **Child had diarrhoea** | |
| No | 7095 (83%) |
| Yes | 1,469 (17%) |
| **Health insurance** | |
| With insurance | 5,036 (59%) |
| Without insurance | 3,527 (41%) |
| **Mother's educational level** | |
| Pre-primary or none | 2,303 (27%) |
| Primary | 1,729 (20%) |
| JSS/JHS/Middle | 3,176 (37%) |
| SSS/SHS/Secondary | 924 (11%) |
| Higher | 432 (5.0%) |
| **Ethnicity** | |
| Akan | 3,959 (46%) |
| Ewe | 877 (10%) |
| Ga/Damgme | 625 (7.3%) |
| Gruma | 375 (4.4%) |
| Grusi | 188 (2.2%) |
| Guan | 382 (4.5%) |
| Mande | 36 (0.4%) |
| Mole Dagbani | 1,455 (17%) |

(*Continued*)

**Table 2.** (Continued)

| Characteristic | N = 8,564[1] |
|---|---|
| Other | 666 (7.8%) |
| **Combined wealth score** | -0.02 (0.91) |
| **Wasting** | |
| Normal | 7,985 (93%) |
| Wasted | 578 (6.8%) |
| **Stunting** | |
| Normal | 7,080 (83%) |
| Stunted | 1,483 (17%) |
| **Underweight** | |
| Normal | 7,514 (88%) |
| Underweight | 1,050 (12%) |

[1] n (%); Mean (SD)

**Underweight.**   Under-five underweight prediction accuracies for all algorithms ranged between 88% and 98% for the test data. The XGBoost and SVM models had the highest value in terms of accuracy, with an optimal balance of sensitivity and specificity (Table 4). Based on AUC values, XGBoost and SVM were the best predictive models followed by LASSO and logistic model (Fig 3).

## The important determinants of childhood undernutrition indicators

As previously described, the XGBoost model was the best for wasting, stunting, and underweight nutritional indicators in terms of accuracy and AUC-ROC characteristics. The top 20 most important variables that contributed to the model's accuracy are presented in Figs 4–6. Top five important variables for wasting were weight of the child, length/height of the child, sex of the child (female), region of residence (Greater Accra), and ethnicity (Gruma). With respect to stunting, the top five variables were age of the child, length/height of the child, sex of the child (female), region of residence (Volta), and the weight of the child. The top five variables for underweight were age of the child, weight of the child, sex of the child (female), ethnicity (Grusi) and region of residence (Brong Ahafo) (Figs 4–6).

## Discussion

Undernutrition in children can result in grave ramifications in their physical and cognitive development. The authors sought to identify the best predictive model and factors associated with undernutrition in children aged five years and younger using the ML approach. In total, 8564 children were included in the final dataset analysis. The average age of the children was 926± 520 days, and most resided in rural areas. Slightly over half of the children were females, with an average current weight of 11.6±3.5kg. Approximately 37% of the mothers had attained juniors high school education with only 16% having attained at least senior high school education. A greater proportion of the children were recruited from the Ashanti Region of Ghana. Approximately 30%, 26%, and 17% of the children were ill with cough, ill with fever, and had diarrhoea, respectively. A greater proportion of 59% had health insurance coverage.

The weighted prevalence rates of stunting, wasting, and underweight were 17%, 6.8%, and 12%, respectively. Most wasted and underweight children were younger on average than normal children, whereas stunted children were, on average, older than normal children [8]. The current weight of the children was significantly lower in all malnourished groups than that in

**Table 3. Weighted prevalence of nutritional outcomes by sociodemographic factors.**

| Characteristic | Wasting | | | Stunting | | | Underweight | | |
|---|---|---|---|---|---|---|---|---|---|
| | Normal | Wasted | p-value[2] | Normal | Stunted | p-value[2] | Normal | Underweight | p-value[2] |
| | N = 7985[1] | N = 578[1] | | N = 7080[1] | N = 1483[1] | | N = 7514[1] | N = 1050[1] | |
| **Age (days)** | 948 (516) | 624 (482) | < 0.001 | 910 (533) | 1004 (450) | < 0.001 | 935 (523) | 864 (497) | 0.002 |
| **Sex** | | | 0.029 | | | 0.007 | | | 0.030 |
| Female | 4080 (51%) | 251 (43%) | | 3665 (52%) | 665 (45%) | | 3861 (51%) | 469 (45%) | |
| Male | 3905 (49%) | 328 (57%) | | 3415 (48%) | 818 (55%) | | 3652 (49%) | 581 (55%) | |
| **Region** | | | 0.3 | | | < 0.001 | | | < 0.001 |
| Ashanti | 1924 (24%) | 131 (23%) | | 1738 (25%) | 318 (21%) | | 1808 (24%) | 247 (24%) | |
| Brong Ahafo | 745 (9.3%) | 57 (9.9%) | | 695 (9.8%) | 108 (7.3%) | | 735 (9.8%) | 67 (6.4%) | |
| Central | 840 (11%) | 67 (12%) | | 743 (10%) | 164 (11%) | | 805 (11%) | 102 (9.7%) | |
| Eastern | 850 (11%) | 40 (7%) | | 753 (11%) | 137 (9.3%) | | 810 (11%) | 81 (7.7%) | |
| Greater Accra | 780 (9.8%) | 46 (8%) | | 726 (10%) | 100 (6.7%) | | 754 (10%) | 72 (6.9%) | |
| Northern | 910 (11%) | 86 (15%) | | 711 (10%) | 284 (19%) | | 814 (11%) | 182 (17%) | |
| Upper East | 259 (3.2%) | 20 (3.4%) | | 230 (3.2%) | 49 (3.3%) | | 237 (3.2%) | 42 (4%) | |
| Upper West | 198 (2.5%) | 12 (2%) | | 179 (2.5%) | 30 (2%) | | 189 (2.5%) | 21 (2%) | |
| Volta | 631 (7.9%) | 55 (9.5%) | | 543 (7.7%) | 144 (9.7%) | | 579 (7.7%) | 108 (10%) | |
| Western | 846 (11%) | 65 (11%) | | 762 (11%) | 149 (10%) | | 782 (10%) | 129 (12%) | |
| **Area** | | | 0.7 | | | < 0.001 | | | 0.13 |
| Rural | 4542 (57%) | 323 (56%) | | 3879 (55%) | 987 (67%) | | 4231 (56%) | 634 (60%) | |
| Urban | 3443 (43%) | 255 (44%) | | 3202 (45%) | 496 (33%) | | 3282 (44%) | 416 (40%) | |
| **Length/Height (cm)** | 86 (14) | 78 (13) | < 0.001 | 87 (14) | 82 (10) | <0.001 | 87 (14) | 80 (11) | < 0.001 |
| **Weight (kg)** | 11.8 (3.4) | 8.1 (2.6) | < 0.001 | 11.8 (3.6) | 10.6 (2.5) | <0.001 | 11.9 (3.5) | 9.1 (2.4) | < 0.001 |
| **Child ill with cough** | | | 0.12 | | | 0.10 | | | 0.2 |
| No | 5583 (70%) | 382 (66%) | | 4969 (70%) | 996 (69%) | | 5264 (70%) | 702 (67%) | |
| Yes | 2402 (30%) | 196 (34%) | | 2111 (30%) | 487 (31%) | | 2250 (30%) | 348 (33%) | |
| **Child ill with fever** | | | 0.005 | | | <0.001 | | | < 0.001 |
| No | 5977 (75%) | 380 (66%) | | 5337 (75%) | 1020 (69%) | | 5654 (75%) | 703 (67%) | |
| Yes | 2008 (25%) | 198 (34%) | | 1743 (25%) | 463 (31%) | | 1860 (25%) | 347 (33%) | |
| **Child with diarrhoea** | | | 0.4 | | | 0.002 | | | 0.001 |
| No | 6628 (83%) | 466 (81%) | | 5931 (84%) | 1164 (78%) | | 6281 (84%) | 814 (78%) | |
| Yes | 1357 (17%) | 112 (19%) | | 1149 (16%) | 319 (22%) | | 1233 (16%) | 236 (22%) | |
| **Health insurance** | | | < 0.001 | | | 0.045 | | | 0.018 |
| With insurance | 4769 (60%) | 268 (46%) | | 4219 (60%) | 818 (55%) | | 4478 (60%) | 558 (53%) | |
| Without insurance | 3216 (40%) | 311 (54%) | | 2861 (40%) | 666 (45%) | | 3035 (40%) | 492 (47%) | |
| **Mother's educational level** | | | 0.092 | | | < 0.001 | | | < 0.001 |
| Pre-primary or none | 2127 (27%) | 176 (30%) | | 1756 (25%) | 547 (37%) | | 1954 (26%) | 349 (33%) | |
| Primary | 1596 (20%) | 134 (23%) | | 1431 (20%) | 298 (20%) | | 1480 (20%) | 249 (24%) | |
| JSS/JHS/Middle | 2994 (37%) | 182 (31%) | | 2696 (38%) | 480 (32%) | | 2861 (38%) | 314 (30%) | |
| SSS/SHS/Secondary | 854 (11%) | 69 (12%) | | 786 (11%) | 137 (9.3%) | | 805 (11%) | 119 (11%) | |
| Higher | 414 (5.2%) | 17 (3%) | | 411 (5.8%) | 21 (1.4%) | | 413 (5.5%) | 18 (1.8%) | |
| **Ethnicity** | | | 0.031 | | | 0.001 | | | 0.048 |
| Akan | 3714 (47%) | 245 (42%) | | 3305 (47%) | 654 (44%) | | 3492 (46%) | 467 (45%) | |
| Ewe | 830 (10%) | 47 (8.1%) | | 756 (11%) | 121 (8.1%) | | 783 (10%) | 94 (8.9%) | |
| Ga/Damgme | 576 (7.2%) | 48 (8.3%) | | 530 (7.5%) | 95 (6.4%) | | 566 (7.5%) | 58 (5.5%) | |
| Gruma | 358 (4.5%) | 17 (2.9%) | | 290 (4.1%) | 85 (5.7%) | | 332 (4.4%) | 42 (4%) | |
| Grusi | 173 (2.2%) | 15 (2.7%) | | 162 (2.3%) | 27 (1.8%) | | 171 (2.3%) | 17 (1.6%) | |
| Guan | 338 (4.2%) | 44 (7.6%) | | 272 (3.8%) | 110 (7.4%) | | 306 (4.1%) | 76 (7.2%) | |

*(Continued)*

**Table 3.** (Continued)

| Characteristic | Wasting | | | Stunting | | | Underweight | | |
| --- | --- | --- | --- | --- | --- | --- | --- | --- | --- |
| | Normal | Wasted | p-value[2] | Normal | Stunted | p-value[2] | Normal | Underweight | p-value[2] |
| | N = 7985[1] | N = 578[1] | | N = 7080[1] | N = 1483[1] | | N = 7514[1] | N = 1050[1] | |
| Mande | 35 (0.4%) | 1 (0.2%) | | 36 (0.5%) | 1 (<0.1%) | | 33 (0.4%) | 3 (0.3%) | |
| Mole Dagbani | 1334 (17%) | 122 (21%) | | 1166 (16%) | 289 (19%) | | 1234 (16%) | 222 (21%) | |
| Others | 627 (7.8%) | 39 (6.8%) | | 564 (8%) | 102 (6.9%) | | 596 (7.9%) | 70 (6.7%) | |
| **Combined wealth score** | -0.02 (0.91) | -0.11 (0.88) | 0.2 | 0.04 (0.93) | -0.31 (0.78) | <0.001 | 0.00 (0.92) | -0.22 (0.79) | < 0.001 |

[1] n (%); Mean (SD)

[2] chi-squared test with Rao & Scott's second-order correction; Wilcoxon rank-sum test for complex survey samples

the normal group. The proportion of children with fever was significantly higher in all malnourished groups than for children in their respective normal groups.

Seven ML algorithms were used to predict undernutrition and to identify factors associated with undernutrition: RF, logistic model, LDA, SVM, ridge regression, XGBoost and LASSO.

The accuracy of all seven ML algorithms with respect to wasting was between 84% and 98%, with a specificity ranging between 83% and 99%, whereas the sensitivity was between 7% and 100% for all algorithms used. The XGBoost model was the most accurate in in predicting wasting and exhibited an optimal balance of sensitivity and specificity. In addition, the accuracy with respect to stunting was between 86% and 98%, with the highest accuracy obtained using the SVM and XGBoost models. The specificity ranged between 92% and 99%, whereas the sensitivity ranged from 27 to 93%, with the SVM and XGBoost models being the most sensitive. Both the SVM and XGBoost models recorded the highest accuracy and showed an optimal balance of sensitivity and specificity to underweight prediction. Previous studies have reported that the accuracy of ML algorithms in predicting undernutrition is between 35.6% and 99.95% [1, 6, 12–14].

The six factors shown to be important for all indicators of undernutrition were age, weight, length/height, sex, region of residence and ethnicity, which was similar to a previous study that employed a logistic regression model to identify these factors [9]. The top five identified factors associated with wasting, stunting, and underweight using the XGBoost model were the weight of the child, age of the child, sex of the child, region of residence and ethnicity. The similarity

**Table 4. Accuracy of Predictive algorithms for child undernutrition indicators on the test data.**

| | LDA | Logistic Regression | SVM | RF | LASSO | Ridge | XGBoost |
| --- | --- | --- | --- | --- | --- | --- | --- |
| **Wasting** | | | | | | | |
| Accuracy | 84% | 95% | 92% | 94% | 96% | 84% | 98% |
| Sensitivity | 100% | 100% | 7% | 12% | 100% | 100% | 90% |
| Specificity | 83% | 94% | 99% | 100% | 96% | 83% | 99% |
| **Stunting** | | | | | | | |
| Accuracy | 86% | 86% | 98% | 88% | 86% | 86% | 98% |
| Sensitivity | 54% | 43% | 93% | 39% | 45% | 27% | 90% |
| Specificity | 92% | 96% | 99% | 99% | 95% | 98% | 99% |
| **Underweight** | | | | | | | |
| Accuracy | 90% | 92% | 98% | 89% | 92% | 88% | 98% |
| Sensitivity | 31% | 51% | 90% | 20% | 55% | 5% | 88% |
| Specificity | 99% | 98% | 99% | 100% | 98% | 100% | 100% |

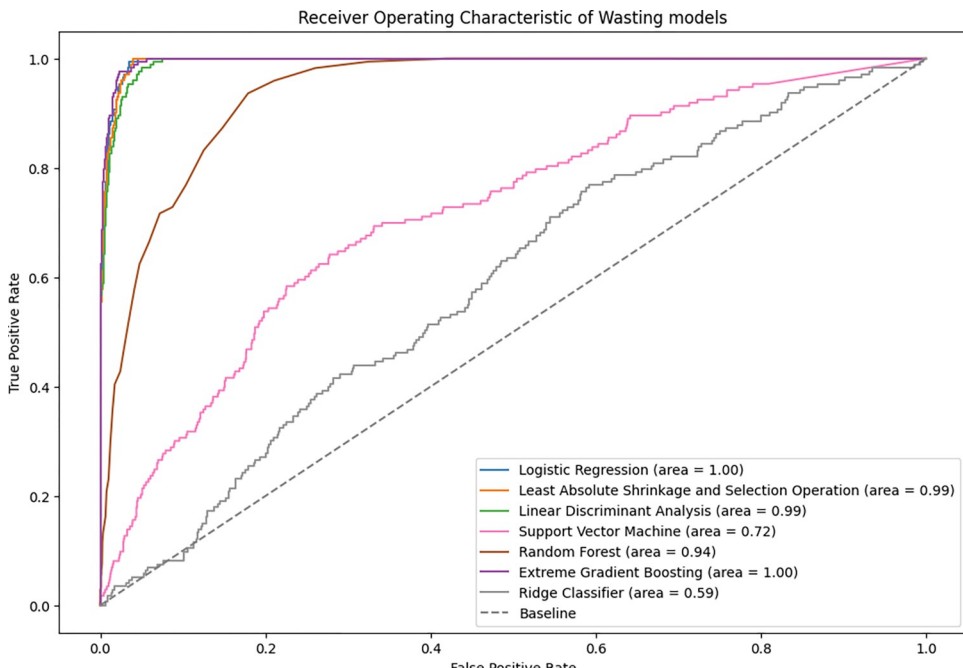

**Fig 1. Receiver operator characteristics on the ML models for wasting test data.**

of the important features was not surprising given that they were generated using an XGBoost model.

XBoost technique has been found to be superior to other machine-learning models in terms of accuracy [24–26]. A study in Ethiopia showed that the XGBoost algorithm is the most

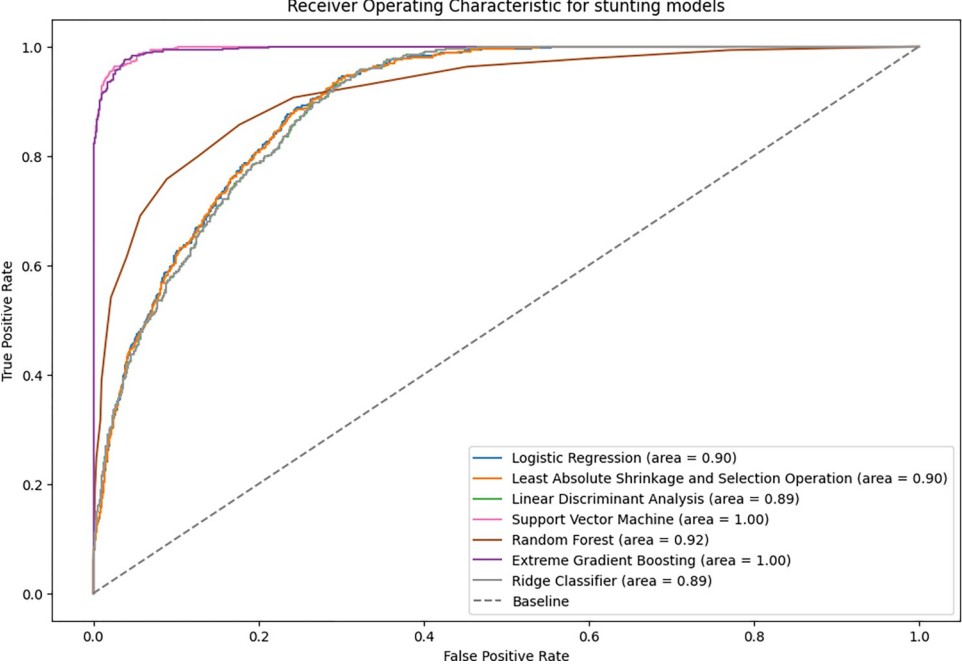

**Fig 2. Receiver operator characteristics on the ML models for stunting on test data.**

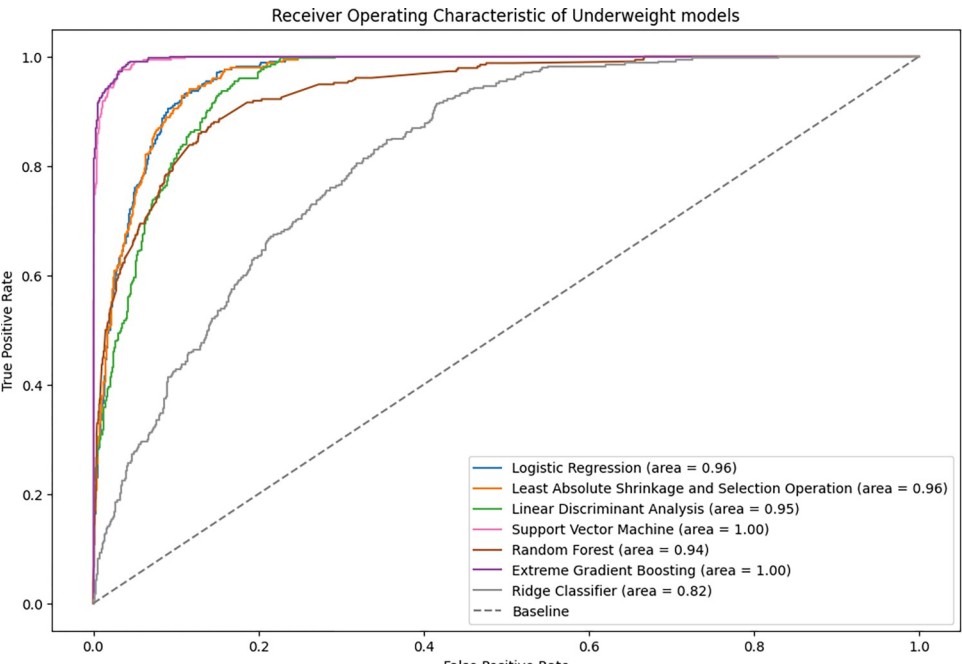

**Fig 3. Receiver operator characteristics on the ML models for underweight on the test data.**

accurate [1]. Another study from Bangladesh reported the highest accuracy when using an artificial neural network [10]. We may attribute the observed superiority of the XGBoost to its ability to leverage the outputs of weak sequential decision trees, where each new tree builds on the weaknesses of the previous trees to make accurate predictions and its ability to effectively handling complex, high-dimensional data for classification [27]. The SVM models followed the XGBoost model with regards to discrimination under the ROC curves in all the three

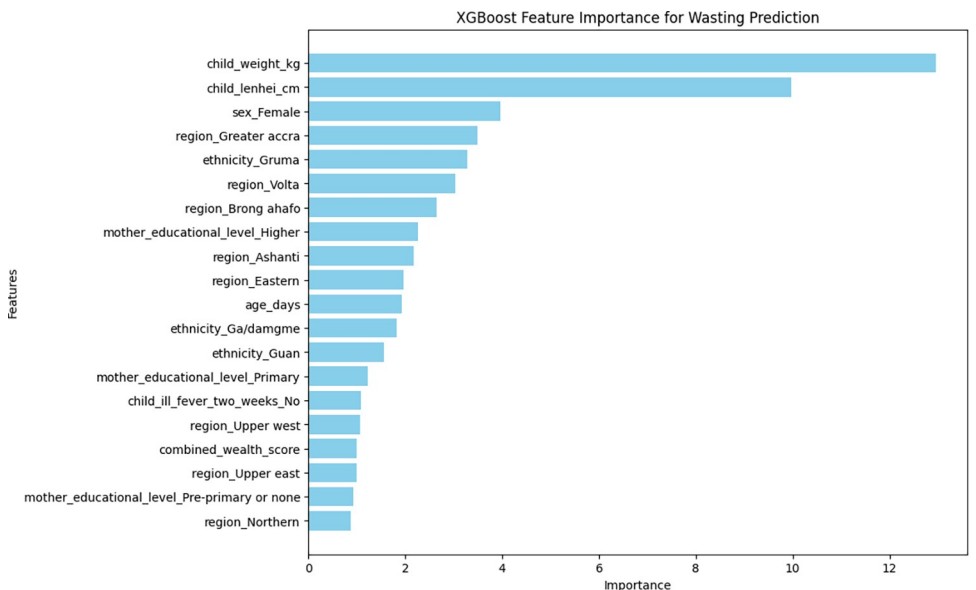

**Fig 4. Top 20 most important variables from the XGBoost model for wasting.**

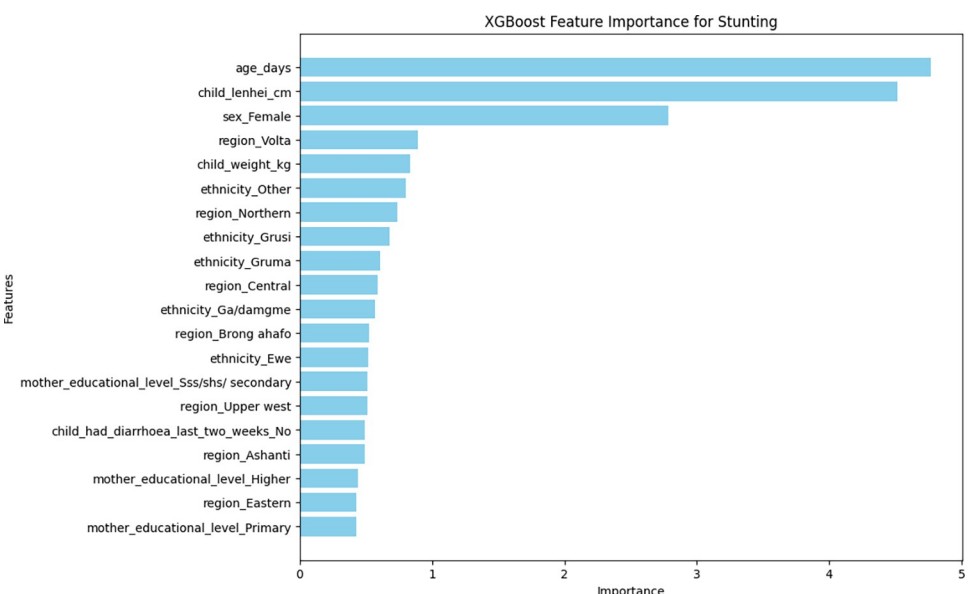

**Fig 5. Top 20 most important variables from the XGBoost model for stunting.**

undernutrition indicators. The strengths of the SVM models may be attributable to the radial basis function (RBF) kernel that enables the SVM to capture relationships between features without explicitly mapping data into high dimension space. A major challenge in comparing the various studies is the evaluation of the different ML algorithms. Regardless, the use of XGBoost model to predict undernutrition among children under five years of age has been demonstrated to be accurate across studies.

Important features associated with wasting in this study were the weight of the child, length/height of the child, sex of the child, region of residence and ethnicity. A similar study in

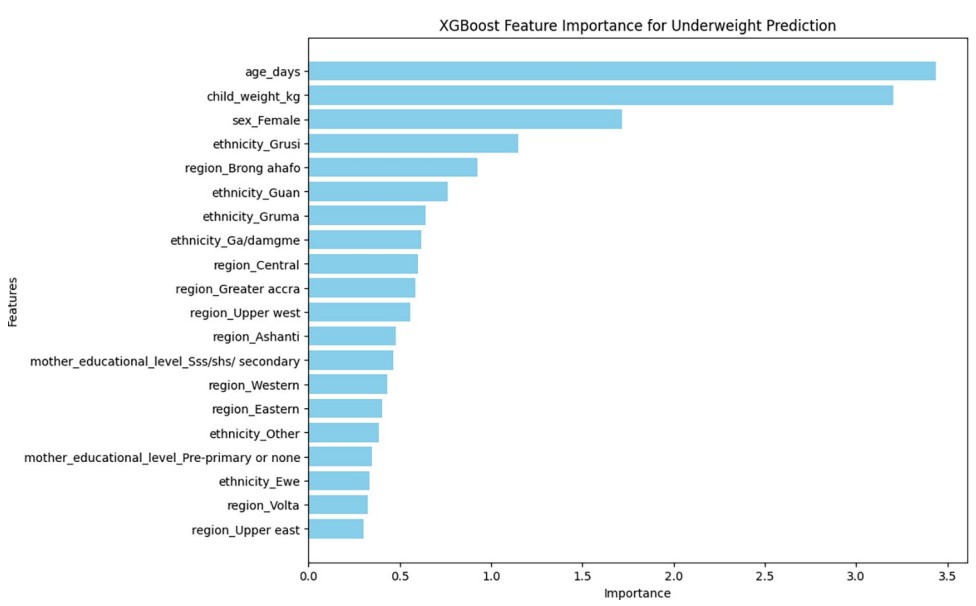

**Fig 6. Top 20 most important variables from the XGBoost model for underweight.**

Ethiopia using the XGBoost algorithm identified children aged > 30 months, wealth index (poorest), time to water intake, ethnicity (Somalia), and small child size as the five main features associated with wasting [1]. Another study in India identified mother's BMI, toilet facility, state, and child's age and religion as important features of wasting [14]. The age of the child is an important factor to consider with respect to wasting among children.

The important features associated with stunting in this study were age of the child, length/height of the child, sex of the child, region of residence, and the weight of the child. A previous study from Ethiopia reported the time to water, child age greater than 30 months, number of children under five in a household, household possession of television, and small child size [1]. Another study from India reported that children's age, toilet facilities, wealth index, mother's education, and breastfeeding duration were important features associated with stunting [14]. Common features identified across studies were weight of the child, age of the child, sex of child, region of residence, and ethnicity.

Important features associated with being underweight in this study were similar to that of stunting and wasting. The important features of a previous study were time to water, lack of education of mothers, small child size, children older than 30 months, and underweight mothers [1]. Common features identified included the age and birth weight of the child. The different features identified for all malnutrition indicators imply that factors associated with undernutrition vary between countries, and as such, the solutions provided should be driven by data to ensure appropriate use of resources.

## Limitations

Factors identified in this study to be associated with undernutrition do not imply causality. Future studies should explore additional factors to help predict malnutrition among children under five years in Ghana. Also, future studies should consider testing the feasibility of machine learning algorithms as potential screening tools for children under five years in Ghana.

## Conclusion

This study highlighted the usefulness of the ML approach in predicting and identifying factors associated with undernutrition in children under five years in Ghana. Weight of the child, age of the child, sex of child, region of residence, and ethnicity are important features associated with undernutrition. Policies targeting the decrease in undernutrition in children should consider these factors. Other factors specific to all nutritional indicators have also been reported to help drive public health actions at the identified factors. The XGBoost models followed by the SVM models were best for predicting wasting, stunting and underweight among children under five years in Ghana. The findings from this study also indicate that different ML models may be useful in predicting undernutrition.

## Author Contributions

**Conceptualization:** Eric Komla Anku, Henry Ofori Duah.

**Data curation:** Eric Komla Anku.

**Formal analysis:** Eric Komla Anku, Henry Ofori Duah.

**Methodology:** Eric Komla Anku, Henry Ofori Duah.

**Writing – original draft:** Eric Komla Anku, Henry Ofori Duah.

**Writing – review & editing:** Eric Komla Anku, Henry Ofori Duah.

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
