## [Decision Letter · Decision Letter 0]

10 Jul 2023

PONE-D-23-10665­­Predicting and Identifying Factors Associated with Undernutrition among Children Under Five Years in Ghana using Machine Learning AlgorithmsPLOS ONE

Dear Dr. Anku,

Thank you for submitting your manuscript to PLOS ONE. After careful consideration, we feel that it has merit but does not fully meet PLOS ONE’s publication criteria as it currently stands. Therefore, we invite you to submit a revised version of the manuscript that addresses the points raised during the review process.

We look forward to receiving your revised manuscript.

Kind regards,

Benojir Ahammed, M.Sc.

Academic Editor

PLOS ONE

Journal Requirements:

Reviewers' comments:

Reviewer's Responses to Questions

**Comments to the Author**

1. Is the manuscript technically sound, and do the data support the conclusions?

Reviewer #1: Yes

Reviewer #2: Yes

2. Has the statistical analysis been performed appropriately and rigorously? 

Reviewer #1: I Don't Know

Reviewer #2: Yes

3. Have the authors made all data underlying the findings in their manuscript fully available?

Reviewer #1: Yes

Reviewer #2: Yes

4. Is the manuscript presented in an intelligible fashion and written in standard English?

Reviewer #1: Yes

Reviewer #2: Yes

5. Review Comments to the Author

Reviewer #1: The manuscript is well written. However the authors should provide justification for selection of the ML models and how was this linked to the type of dataset analysed/trained.

Lack of sensitivity of many models should be explained by doing indepth discussion of the performance of each model. The authors discuss more the output of the models but little on the selected models.

Why size of the dataset is considered as a limitation while it was possible to use a large dataset?

Reviewer #2: -The present paper has tried to determine the predictors of malnutrition in children using various statistical methods. However, the reason for using several methods for a time point is not clear.

A study with the following title was published by Mercedes de Onis [et al] in 2004.

“Methodology for estimating regional and global trends of child malnutrition”

The paper described the methodology developed by the World Health Organization (WHO) to derive global and regional trends of child stunting and underweight, and reports trends in prevalence and numbers affected for 1990–2005.

The proposed method in that study can be used to analyze trends and predictors over the time.

It is recommended to refer to the previous method and mention the reasons of the need of other methods, especially considering that the methods proposed in this paper have very low sensitivities. - Page 18: It seems there was a mistake between sensitivity and specificity.

Despite showing high levels of specificity (99% to 100%) for predicting under-five wasting, all models had very poor sensitivity (0% to 10%) (Table 4, Figure 1).

6. PLOS authors have the option to publish the peer review history of their article (what does this mean?). If published, this will include your full peer review and any attached files.

Reviewer #1: No

Reviewer #2: **Yes: **Bahareh Nikooyeh

---

## [Author Response · Author response to Decision Letter 0]

24 Aug 2023

We thank the editor and reviewers for taking their time to review our paper and the constructive comments they have offered. We have responded to all feedback from reviewers point by point. Changes have been made in the file named “Revised Manuscript with Track Changes” and an unmarked version in the file named “Manuscript”, and we have provided a summary of our feedback below: 

Response to Academic Editor

Comment 1: Please ensure that your manuscript meets PLOS ONE's style requirements, including those for file naming. The PLOS ONE style templates can be found at 

Response: We have made changes to the manuscript to meet the requirements stated in the attached document and the file names.

Comment 2: Please provide additional details regarding participant consent. In the ethics statement in the Methods and online submission information, please ensure that you have specified what type you obtained (for instance, written or verbal, and if verbal, how it was documented and witnessed). If your study included minors, state whether you obtained consent from parents or guardians. If the need for consent was waived by the ethics committee, please include this information.

Response: Information regarding ethics has been updated in the ethical consideration section.

Comment 3: Please note that PLOS ONE has specific guidelines on code sharing for submissions in which author-generated code underpins the findings in the manuscript. In these cases, all author-generated code must be made available without restrictions upon publication of the work. Please review our guidelines at https://journals.plos.org/plosone/s/materials-and-software-sharing#loc-sharing-code and ensure that your code is shared in a way that follows best practice and facilitates reproducibility and reuse.

Response: The code used for analysis has been made available, and the information is updated in the data availability statement.

Comment 4: In your Data Availability statement, you have not specified where the minimal data set underlying the results described in your manuscript can be found. PLOS defines a study's minimal data set as the underlying data used to reach the conclusions drawn in the manuscript and any additional data required to replicate the reported study findings in their entirety. All PLOS journals require that the minimal data set be made fully available. For more information about our data policy, please see http://journals.plos.org/plosone/s/data-availability.

Response: 

Data availability statement

This study was based on a publicly available dataset with no personal identifiers, and is freely available upon request from the DHS program (DHS). The code used for the analysis can be found in the repository.

Comment 5: Please review your reference list to ensure that it is complete and correct. If you have cited papers that have been retracted, please include the rationale for doing so in the manuscript text, or remove these references and replace them with relevant current references. Any changes to the reference list should be mentioned in the rebuttal letter that accompanies your revised manuscript. If you need to cite a retracted article, indicate the article’s retracted status in the References list and also include a citation and full reference for the retraction notice.

Response: References have been reviewed.

Response to Reviewer 1

Comment: The manuscript is well written. However, the authors should provide justification for selection of the ML models and how was this linked to the type of dataset analysed/trained.

Response: Thank you very much for your review. The models selected are classification machine learning models and they were selected because of the outcome of interest which was binary

Comment: Lack of sensitivity of many models should be explained by doing in-depth discussion of the performance of each model. The authors discuss more the output of the models but little on the selected models. 

Response: We oversampled the training dataset to deal with class imbalance of the minority class which improved the sensitivity of the models. We have also added discussion point on our explanation for the superiority in discrimination observed for the Random Forest and the SVM models. 

Comment: Why size of the dataset is considered as a limitation while it was possible to use a large dataset?

Response: Limitation cited regarding the size of dataset has been deleted because it is no longer pertinent after having address the class imbalance problem.

Response to Reviewer 2

Comment: The present paper has tried to determine the predictors of malnutrition in children using various statistical methods. However, the reason for using several methods for a time point is not clear. A study with the following title was published by Mercedes de Onis [et al] in 2004.

“Methodology for estimating regional and global trends of child malnutrition” The paper described the methodology developed by the World Health Organization (WHO) to derive global and regional trends of child stunting and underweight, and reports trends in prevalence and numbers affected for 1990–2005.

The proposed method in that study can be used to analyze trends and predictors over the time. It is recommended to refer to the previous method and mention the reasons of the need of other methods, especially considering that the methods proposed in this paper have very low sensitivities.). 

Response: Thank you very much for your feedback and also for sharing the study by Mercedes de Onis [et al] in 2004. We reviewed the study and found that the methods proposed in the study have different objectives compared to our study. Our study aimed at predicted stunting, wasting and underweight and not the trends and prevalence over time as proposed in the stated study. 

Comment: - Page 18: It seems there was a mistake between sensitivity and specificity. Despite showing high levels of specificity (99% to 100%) for predicting under-five wasting, all models had very poor sensitivity (0% to 10%) (Table 4, Figure 1 

Response: We have also looked at the error pointed out from the manuscript and made changes accordingly. Once again, we are grateful for your time and feedback.

---

## [Decision Letter · Decision Letter 1]

18 Dec 2023

­­Predicting and Identifying Factors Associated with Undernutrition among Children Under Five Years in Ghana using Machine Learning Algorithms

PONE-D-23-10665R1

Dear Dr. %Anku%,

We’re pleased to inform you that your manuscript has been judged scientifically suitable for publication and will be formally accepted for publication once it meets all outstanding technical requirements.

Kind regards,

Benojir Ahammed, M.Sc.

Academic Editor

PLOS ONE

Additional Editor Comments (optional):

Reviewers' comments:

Reviewer's Responses to Questions

**Comments to the Author**

1. If the authors have adequately addressed your comments raised in a previous round of review and you feel that this manuscript is now acceptable for publication, you may indicate that here to bypass the “Comments to the Author” section, enter your conflict of interest statement in the “Confidential to Editor” section, and submit your "Accept" recommendation.

Reviewer #1: All comments have been addressed

2. Is the manuscript technically sound, and do the data support the conclusions?

Reviewer #1: Yes

3. Has the statistical analysis been performed appropriately and rigorously? 

Reviewer #1: Yes

4. Have the authors made all data underlying the findings in their manuscript fully available?

Reviewer #1: No

5. Is the manuscript presented in an intelligible fashion and written in standard English?

Reviewer #1: Yes

6. Review Comments to the Author

Reviewer #1: The authors have effectively addressed the feedback concerning the methodology, results, and discussion sections, and implementing the required revisions to enhance the manuscript.

7. PLOS authors have the option to publish the peer review history of their article (what does this mean?). If published, this will include your full peer review and any attached files.

Reviewer #1: No

---

## [Editor Report · Acceptance letter]

5 Feb 2024

PONE-D-23-10665R1 

PLOS ONE

Dear Dr. Anku, 

I'm pleased to inform you that your manuscript has been deemed suitable for publication in PLOS ONE. Congratulations! Your manuscript is now being handed over to our production team.

Kind regards, 

on behalf of

Mr. Benojir Ahammed 

Academic Editor

PLOS ONE